# In Vitro Assessment of Berberine against *Ichthyophthirius multifiliis* in Goldfish

**DOI:** 10.3390/pathogens11101207

**Published:** 2022-10-20

**Authors:** Ke Huang, Guangran Hu, Runqiu Wang, Qingwen Zeng, Wenxiang Li, Hong Zou, Shangong Wu, Guitang Wang, Ming Li

**Affiliations:** 1State Key Laboratory of Freshwater Ecology and Biotechnology, and Key Laboratory of Aquaculture Disease Control, Ministry of Agriculture, Institute of Hydrobiology, Chinese Academy of Sciences, Wuhan 430072, China; 2University of Chinese Academy of Sciences, Beijing 100049, China; 3Hubei Key Laboratory of Three Gorges Project for Conservation of Fishes, Chinese Sturgeon Research Institute, China Three Gorges Corporation, Yichang 443100, China

**Keywords:** *Ichthyophthirius multifiliis*, berberine, theronts, tomonts, acute toxicity

## Abstract

*Ichthyophthirius multifiliis* is a pathogenic ciliate parasite, which infects almost all freshwater teleost fish and leads to significant economic losses. The present study aimed to evaluate the acute toxicity of berberine to the free-living stages of *I. multifiliis*, that is, theronts and tomonts. Our results indicated that 99.30% of *I. multifiliis* theronts were killed by a concentration of 15 mg/L berberine during the 4 h exposure time, while berberine had no effect on protomonts. Nevertheless, berberine at a concentration of 5 mg/L could effectively reduce the release of theronts from tomonts treated for 4 h. Additionally, according to the transmission electron microscopy results, berberine at 15 mg/L could strongly change the shape of protomonts, destroy their organelles, and significantly decrease the number of ribosomes. The median lethal concentration (LC_50_) of berberine for goldfish at 96 h was 528.44 mg/L, which was almost 67 times the median effective concentration (EC_50_) of berberine for killing theronts (7.86 mg/L). The results demonstrated that berberine could be an effective and safe potential parasiticide for killing *I. multifiliis*.

## 1. Introduction

*Ichthyophthirius multifiliis* Fouquet, 1876, is one of the most harmful parasitic pathogens in freshwater teleosts, with low host specificity [1,2]. It causes ichthyophthiriasis, commonly known as the “white spot disease”, which can cause great economic losses to the global aquaculture industry [1,2,3]. It has four developmental stages (Figure 1): a parasitic trophont, a free-swimming protomont, a reproductive tomont, and an infective theront [3,4]. When the trophont matures, it naturally sheds itself from the host and becomes a free-swimming protomont; when the protomont attaches itself to a flat surface, it soon transforms into a tomont. The tomont then undergoes binary fission, producing hundreds of daughter cells called tomites, which further form infectious theronts. Theronts invade the epidermis of susceptible fish, and then rapidly transform into trophonts to complete their entire life cycle [2,4,5,6].

The prevention and control of ichthyophthiriasis have long been a key issue to fish parasite disease. Malachite green, as the most effective parasiticide to treat ichthyophthiriasis [7], has been banned in aquaculture practice because of its teratogenic, genotoxic, and carcinogenic effects [8,9,10]. Meanwhile, other chemicals, such as formalin [11], copper sulphate [12], potassium permanganate [13], and so on, have raised some environmental concerns [6]. Therefore, it is necessary to search for efficient, alternative, and environmentally friendly products to control *I. multifiliis* infection.

Herbal medicines have the characteristics of easy biodegradation and a low environmental hazard, which, in general, makes them environmentally friendly [14]. Multiple studies have shown that some plants possess effective parasiticide properties, and that some plant extracts show high efficacy against the free-swimming stages of *I. multifiliis* [5,15,16,17,18,19,20]. Berberine (BBR, C_20_H_19_NO_5_, MW 336.37) is an isoquinoline alkaloid and a positively charged particle isolated from several species of plants popular in Chinese herbal medicines, but most commonly from *Coptis chinensis*. Owing to its wide range of sources, low cost, and low toxicity, there have been many studies on the pharmacological effects of berberine, including the antibacterial [21], antineoplastic [22], and antiparasitic [23] properties. Given this background, we speculated that berberine may be an effective treatment against the free-living *I. multifiliis* (theronts and tomonts) in fish, but this hypothesis had not been tested before. In the present study, we used berberine and *I. multifiliis* to determine the efficacy and antiparasitic mechanism of berberine in vitro against the free-living *I. multifiliis* (theronts and protomonts).

## 2. Materials and Methods

### 2.1. Fish

Goldfish (*Carassius auratus*) weighing 14.2 ± 1.4 g (mean ± SD) were obtained from a local ornamental fish shop in Wuhan, China. All fish were distributed into several 30 L transparent tanks, and the tanks were kept under a flow system with dissolved oxygen at 6.0–8.0 mg/L, pH at 7.4 ± 0.2, a constant temperature of 23.0 ± 0.5 °C, and a light:dark period of 12:12 h. The fish were fed 1% body weight once a day with commercial fish pellet feedstuffs (Wuhan, China). These goldfish were partly used for propagation of *I. multifiliis* to obtain experimental protomonts and theronts, and partly used for the testing of 96 h acute toxicity of berberine to goldfish. 

### 2.2. Parasites

*Ichthyophthirius multifiliis* was isolated from the goldfish. Several heavily infected fish were placed into 500 mL of aerated water for 30 min, which caused mature trophonts to fall off the fish. The unsettled protomonts were either directly randomly assigned into 24-well microtiter plates for medicinal tests or collected into 35 mm Petri dishes containing distilled water to hatch into theronts in a 23.0 ± 0.5 °C incubator.

### 2.3. Berberine

Berberine with a purity greater than 97% (HPLC) was purchased from Macklin (Shanghai, China). Before the experiments, berberine was mixed with distilled water to formulate into a series of stock solutions with different concentrations of 2, 5, 10, 15, 20, 30, and 40 mg/L, stored at 4 °C in a refrigerator, and settled in room temperature before use.

### 2.4. Effect of Berberine on I. multifiliis Theronts

In vitro studies that were conducted to determine the antiprotozoal activity of berberine against *I. multifiliis* theronts were adapted from the previously devised method [12,15,24]. Theront concentration was determined by taking 100 μL droplets from the theront suspension and calculating the average of five counts via a Sedgewick Rafter counting chamber and a microscope (40× magnification) [24,25]. Then, 100 μL of theront solution per well was placed into a 96-well microtiter plate and exposed to 100 μL berberine solutions, so that the final concentrations in the 96-well plate were 0, 1.0, 2.5, 5.0, 7.5, 10.0, 15.0, and 20.0 mg/L. Acute toxicity was assessed by the mortality of theronts 4 h after the treatment; theronts with an absence of motility and abnormal morphology were considered dead. The trial was conducted at 23.0 ± 0.5 °C and repeated three times.

### 2.5. Effect of Berberine on I. multifiliis Protomonts

According to the method described above, an in vitro study was designed to assess the efficacy of berberine against *I. multifiliis* tomonts. Protomonts that had discarded fish mucus were distributed into a 24-well microtiter plate, at the concentration of 30 protomonts (with 1000 μL of distilled water) per well. Beforehand, 1000 μL of prepared stock solutions was added to each well, so that the final concentrations of each well were 0, 1.0, 2.5, 5.0, 7.5, 10.0, 15.0, and 20.0 mg/L. Distilled water was used for the control group. After 4 h, the acute toxicity of berberine to protomonts was directly assessed by the number of dead protomonts. After 20 h of treatment, the percentage of theronts released from tomonts was assessed via microscopic examination. The mortality of protomonts was determined by their cytoplasmic immobility and abnormal morphology. The experiment was conducted at 23.0 ± 0.5 °C, and all groups were tested three times.

### 2.6. Ultrastructural Analysis 

Berberine-treated for 4 h and untreated (control group) *Ichthyophthirius multifiliis* protomonts were fixed with 0.1 M phosphate-buffered 2.5% glutaraldehyde (pH 7.2) over 12 h at 4 °C. The parasite samples were processed according to the method described by Ozaki et al. [26]. Ultrastructural analysis was performed using a Hitachi 7700 transmission electron microscope (Tokyo, Japan) at 80.0 kV acceleration voltage.

### 2.7. Acute Toxicity of Berberine to Goldfish

The acute toxicity of berberine to goldfish was determined by the method of aqueous static renewal 96 h bioassays. Goldfish were divided into five aquariums (10 fish/aquarium), and a series of berberine solutions with different concentrations of 0 (control group), 300, 400, 500, and 600 mg/L were prepared for goldfish aqueous exposures. Mortality observations and recordings of goldfish in the aquariums were taken every 24 h. The fish were not fed during the exposures.

### 2.8. Statistical Analysis

Statistical analyses were performed using SPSS 18.0 (SPSS Inc., Chicago, IL, USA) and GraphPad Prism 8.0 (Graphpad Inc., San Diego, CA, USA). The data of in vitro trials were analyzed using a nonparametric method (the Kruskal–Wallis test), and significant differences between treatment and control groups were estimated using a one-way ANOVA (significant differences *p* < 0.05), followed by Dunnett’s test for multiple comparisons. The EC_50_ values and their 95% confidence intervals were calculated by the GraphPad Prism 8.0 dose procedure, and the LC_50_ values and their 95% confidence intervals were calculated by the SPSS 18.0 probit procedure. 

## 3. Results

### 3.1. Acute Toxicity of Berberine to I. multifiliis Theronts and Protomonts

The mortality of theronts at a concentration of 0 mg/L was 18.43 ± 3.97% (Table 1). The mortality of theronts at a concentration of 5 mg/L was significantly higher (43.10 ± 3.35%), and the administration of 15 mg/L of berberine for 4 h killed all *I. multifiliis* theronts (100%).

The efficacy of berberine against protomonts was low: at 20 mg/L berberine concentration, their mortality was only 4.44 ± 1.92% (Table 1). However, berberine was effective at obstructing the generation of tomonts: at a low concentration of 2.5 mg/L, only 37.51 ± 3.81% theronts were released from tomonts, and at concentrations greater than 5 mg/L, no tomonts transformed into theronts (Table 1). 

### 3.2. Ultrastructural Observation

The *I. multifiliis* protomonts in the control group were smooth and round (Figure 2A). The mitochondria were spherical to ovoid in shape (0.77–1.09 μm in length and 0.27–0.44 μm in width), and the tubular structures of their inner membrane were intact (Figure 2B). Ribosomes were scattered around the cytoplasm (Figure 2C).

After treatment with berberine, *I. multifiliis* protomonts exhibited several signs of damage. The cells were deformed and showed cytoplasmic vacuolation (Figure 2D), the inner membrane of the mitochondria was destroyed (Figure 2E), and the mitochondria exhibited considerable swelling after 4 h of treatment (0.62–0.89 μm in length and 0.44–0.63 μm in width). The number of ribosomes was decreased significantly in the cytoplasm (Figure 2F). Moreover, a great deal of organelle fragments were observable in the cytoplasm of berberine-treated protomonts (Figure 2F).

### 3.3. Acute Toxicity of Berberine to Goldfish

The goldfish exposed to various concentrations of berberine showed different responses to treatment within 96 h post-treatment (Table 2). Goldfish did not die under the treatment with berberine at concentrations of 0 (control) or 300 mg/L. The fish treated with berberine at 400 or 500 mg/L showed a mortality rate of 30%, and 70% of goldfish died when treated with 600 mg/L of berberine within 96 h (Table 2). The 96 h LC_50_ value for berberine in goldfish was 528.44 mg/L. (Table 3).

## 4. Discussion

The disease caused by *I. multifiliis* probably brings about more damage to freshwater fish than any other eukaryotic pathogen [2,3]. There are two main strategies for the treatment of the disease: one is aiming at the parasitic trophont stage within the host′s epithelium, while the other is targeting the external stages where the parasite is free-living [2,4]. However, the trophonts in the epithelium are protected both by the mucus secreted by the fish and the proliferation of epithelial tissue, so it is difficult for drugs to kill trophonts through the epithelial tissue.

Straus and Griffin [24] argued that killing infectious theronts could prevent the disease from spreading to other fish. Several studies reported that theronts are more sensitive than other stages of *I. multifiliis* to plant extracts [5,16,20] and chemicals [11,15]. Killing theronts can effectively prevent fish from being infected by *I. multifiliis*. It is therefore of high priority to access the antiprotozoal activity of berberine against *I. multifiliis* theronts. Our results showed that berberine could kill almost all theronts at a concentration of 15 mg/L in 4 h, and berberine concentrations of 7.5 mg/L and more had over 60% efficiency in killing *I. multifiliis* theronts. In comparison, it took 4 h to kill all theronts using commercially available copper naphthenate at a concentration of 8 mg/L (organic copper content = 2.4%) [27]. The concentrations of ethanol extracts of *Zingiber officinale* and *Cynanchum atratum* needed to achieve 100% theronts mortality were 8 mg/L and 16.0 mg/L, respectively [28]. The concentration of berberine needed to achieve 100% theront mortality at the 4 h time point was much lower than that of garlic extract, which required 62.5 mg/L for 15 h [15]. However, 0.6 mg/L of magnolol managed to eliminate all theronts after 4 h [29]. Therefore, berberine could effectively eliminate theronts in water.

Although tomonts do not directly invade fish, they can produce a large number of infective theronts [2]. Therefore, it is important to stop theront infection by preventing the productive tomonts. At a concentration of 20 mg/L berberine could kill all theronts quickly, but only killed 4.44% of protomonts, which indicated that protomonts are more resistant to berberine than theronts. This is similar to results obtained using other plant extracts [20,30]. Even so, at a concentration of 5 mg/L for 4 h, berberine could completely prevent tomonts from releasing theronts. This suggests that berberine may effectively block the life cycle of *I. multifiliis* even at low concentrations, and thus significantly decrease the re-infection with theronts and increase the survival rate of infected fish.

Previous studies have shown that Chinese herbal extracts not only have a good in vitro anti-*I. multifiliis* effect, but also have low toxicity to fish, whose LC_50_ is generally several to dozens of times higher than that of the EC_50_ against theronts. For example, psoralen at a concentration of 0.8 mg/L can kill all *I. multifiliis* theronts within 4 h, but the LC_50_ at 96 h is 5.6 mg/L [31]. For magnolol, the 96 h LC_50_ value for goldfish is 6.02 mg/L, but only 0.6 mg/L of magnolol can kill all theronts in 4 h [29]. In our study, the LC_50_ (528.44 mg/L) for goldfish was about 67 times larger than the EC_50_ (7.86 mg/L) for killing theronts, and 192 times higher than the EC_50_ (2.75 mg/L) for inhibiting the development of cysts into theronts (Table 3). The results showed that berberine is an effective and safe drug against *I. multifiliis* in vitro.

Furthermore, our TEM results demonstrated that berberine treatment resulted in the damage of organelles and cytomembranes, reflected in the reduction in the number of ribosomes, the rupturing of the inner membrane of mitochondria, and mitochondrial swelling. This is similar to the results of Qu′s and Ruider′s studies [7,32]. Ribosomes are critical in protein production in all cells, and the *I. multifiliis* is no exception. Their major function is to promote the translational process, and mutations or alterations of ribosomal proteins cause abnormal cell proliferation [33,34]. Mohammad et al. [35] reported that one of the targets of aminoglycosides was the ribosomes of Leishmania. Therefore, we suspected that the antiparasitic function of berberine may relate to the decrease in the number of ribosomes in *I. multifiliis*. Mitochondrial changes may also be directly involved in the cell death process of *I. multifiliis*. Some studies indicated that berberine could inhibit various human tumor cells by inducing mitochondrial apoptosis [36,37]. In addition, several reports provided evidence that mitochondria are one of the primary cellular targets of berberine in human tumor cells [36,37,38]. Our results suggested that berberine causes structural alterations in parasite cells, particularly severe disruption to the inner membrane of mitochondria. These effects are similar to those achieved by the treatment with malachite green [7,32] and 10-gingerol [39]. In Fu’s study, 10-gingerol reduced ATP ase activities, leading to the accumulation of free radicals, which further led to the destruction of cell membranes and organelles. [39] The mechanism by which berberine kills *I. multifiliis* may be similar to that of 10-gingerol, but more studies are needed to test this hypothesis.

## Figures and Tables

**Figure 1 pathogens-11-01207-f001:**
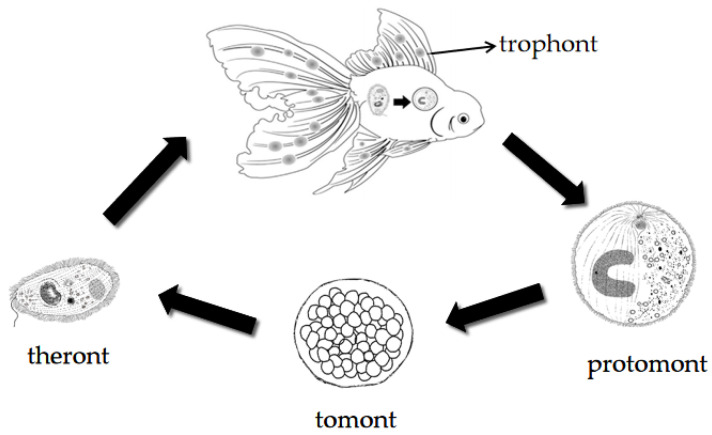
The life cycle of *Ichthyophthirius multifiliis*: (1) The infectious theront invades the epidermis of fish and grows into a trophont. (2) Mature trophont leaves the host to become a free-swimming protomont. (3) Protomont secretes a cyst wall to become a tomont. (4) Tomont undergoes binary fission, producing large numbers of theronts to invade fish again (cited from Li et al. [4]).

**Figure 2 pathogens-11-01207-f002:**
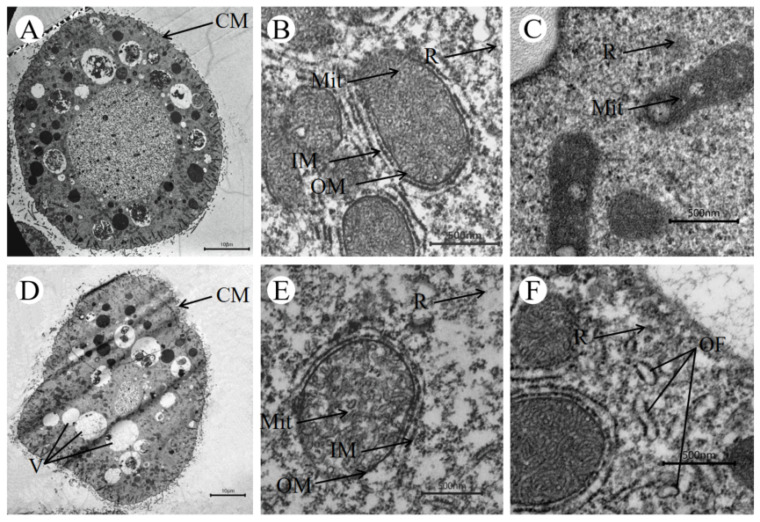
Transmission electron micrographs of untreated (**A**) and 4 h treated (**D**) *I. multifiliis* trophonts. (**A**) The overall structure of untreated trophont. Scale bar = 10 µm (**B**) Normal mitochondria and large numbers of ribosomes in the control group. Scale bar = 500 nm. (**C**) Large numbers of ribosomes in the control group. Scale bar = 500 nm. (**D**) The overall structure of a trophont treated for 4 h with berberine. Scale bar = 10 µm. (**E**) Destruction of mitochondrial inner membrane and reduction in the number of ribosomes in the berberine-treated group. Scale bar = 500 nm. (**F**) Reduction in the number of ribosomes and organelle fragments in the berberine-treated group. Scale bar = 500 nm. Abbreviations: CM, cell membrane; V, vacuole; Mit, mitochondrion; IM, inner membrane; OM, outer membrane; R, ribosome; OF, organelle fragment.

**Table 1 pathogens-11-01207-t001:** Efficacy of berberine against *I. multifiliis* theronts and tomonts.

Concentration (mg/L)	Theronts	Tomonts	
	MT (%)	MT (%)	PRT (%)
0	18.43 ± 3.97 ^a^	0.00 ± 0.00 ^a^	100.00 ± 0.00 ^a^
1	23.11 ± 17.02 ^a^	0.00 ± 0.00 ^a^	93.33 ± 3.33 ^b^
2.5	30.82 ± 18.36 ^ab^	0.00 ± 0.00 ^a^	37.51 ± 3.81 ^c^
5	43.10 ± 3.35 ^b^	0.00 ± 0.00 ^a^	0.00 ± 0.00 ^d^
7.5	65.41 ± 3.48 ^c^	0.00 ± 0.00 ^a^	0.00 ± 0.00 ^d^
10	72.66 ± 7.05 ^c^	0.00 ± 0.00 ^a^	0.00 ± 0.00 ^d^
15	99.30 ± 1.21 ^d^	0.00 ± 0.00 ^a^	0.00 ± 0.00 ^d^
20	100.00 ± 0.00 ^d^	4.44 ± 1.92 ^b^	0.00 ± 0.00 ^d^

MT: mortality of theronts and tomonts after 4 h. PRT: percentage of theronts released from tomonts after 20 h. The mortality data are given as a mean ± standard deviation. In the same column, the values with different lowercase letters (a, b, c, d) are significantly different (*p* < 0.05).

**Table 2 pathogens-11-01207-t002:** Acute toxicity of berberine to goldfish in aqueous static renewal 96 h bioassays.

Concentration (mg/L)	Total No. Tested	No. of Dead				Survival (%)
		24 h	48 h	72 h	96 h	
Control	10	0	0	0	0	100
300	10	0	0	0	0	100
400	10	0	0	3	0	70
500	10	0	3	0	0	70
600	10	4	0	2	1	30

**Table 3 pathogens-11-01207-t003:** In vitro anti-theront efficacy and acute toxicity of berberine to healthy goldfish.

In Vitro Anti-Theront Efficacy at 4 h		Acute Toxicity to Goldfish at 96 h	
EC_50_	95% Confidence Interval	LC_50_	95% Confidence Interval
7.86	6.61–14.92	528.44	459.07–713.70

## Data Availability

Not applicable.

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
