# Peer review of "In Vitro Assessment of Berberine against Ichthyophthirius multifiliis in Goldfish"

_pathogens, 2022, doi:10.3390/pathogens11101207_

Round 1

Reviewer 1 Report

Abstract: line 14 instead free-living periods say free-living life stages.

Introduction Line 28: Do you mean low host-specificity?

Methods: Line 67: are these the same goldfish as in section 2.1?

Line 73: change to Macklin (Shanghai, China)

Line 78 in vitro italicised.

Results: Line 162 is there mortality at 300 mg/L?

Line 168: in vitro italicised

Reviewer 2 Report

The article is interesting. The authors describe the effect of berberine against the parasite Ichthyophthirius multifiliis. A very important observation is that berberine at 15 mg/L changes the shape of protomonts, destroys their organelles, and decreases the number of ribosomes.

I suggest some corrections:

1. In the Materials and Methods should be more details, including the composition of the feed, pH of the water. Why was used distilled water, instead of 0.9% NaCl solution or the addition of horse serum?

2. Please add in the Introduction, the figure with a life cycle of Ichthyophthirius multifiliis, for better understanding by readers not knowing this parasite.
